# Neural Surfel Reconstruction: Addressing Loop Closure Challenges in Large-Scale 3D Neural Scene Mapping

**DOI:** 10.3390/s24216919

**Published:** 2024-10-28

**Authors:** Jiadi Cui, Jiajie Zhang, Laurent Kneip, Sören Schwertfeger

**Affiliations:** Key Laboratory of Intelligent Perception and Human-Machine Collaboration, ShanghaiTech University, Ministry of Education, Shanghai 201210, China; cuijd@shanghaitech.edu.cn (J.C.); zhangjj2023@shanghaitech.edu.cn (J.Z.)

**Keywords:** 3D scene reconstruction, large-scale reconstruction, surfel, neural representation, loop closure

## Abstract

Efficiently reconstructing complex and intricate surfaces at scale remains a significant challenge in 3D surface reconstruction. Recently, implicit neural representations have become a popular topic in 3D surface reconstruction. However, how to handle loop closure and bundle adjustment is a tricky problem for neural methods, because they learn the neural parameters globally. We present an algorithm that leverages the concept of surfels and expands relevant definitions to address such challenges. By integrating neural descriptors with surfels and framing surfel association as a deformation graph optimization problem, our method is able to effectively perform loop closure detection and loop correction in challenging scenarios. Furthermore, the surfel-level representation simplifies the complexity of 3D neural reconstruction. Meanwhile, the binding of neural descriptors to corresponding surfels produces a dense volumetric signed distance function (SDF), enabling the mesh reconstruction. Our approach demonstrates a significant improvement in reconstruction accuracy, reducing the average error by 16.9% compared to previous methods, while also generating modeling files that are up to 90% smaller than those produced by traditional methods.

## 1. Introduction

There are already many methods that have been successful in generating object-level 3D meshes, but creating meshes for large scenes still requires manual modeling, which is very time-consuming and labor-intensive. This is because large scenes often have trajectory drift issues, making it challenging for various representations to aim for realistic modeling.

A naive and intuitive 3D surface reconstruction approach [1] is to paste many “papiers” (a 2D surface, potentially bend, in the 3D scene) in the environment to be modeled, and preserve all information about these papiers’ positions, shapes, etc., to obtain the reconstruction result. As shown in Figure 1, this papier-modeling method is actually very effective and flexible, but how to represent these papiers digitally is a problem.

Surfels [2,3,4], short for surface elements, having similar properties, are small circular disks defined by a 3D position and normal vector. They provide a continuous and versatile surface representation for effective 3D modeling. By densely covering object surfaces with surfels, we can approximate detailed geometry at arbitrary resolution. However, traditional definitions of surfels have many limitations. For example, they lack continuity since each surfel is independent, making it difficult to represent completely watertight surfaces. The fixed size of standard surfels also limits precision, as their resolution is constrained. There is a strong trade-off between accuracy and storage space. Larger surfels can cover more area but sacrifice detail, while smaller ones have high costs.

Neural representation [5,6] will lend the flexibility needed for continuity and completeness of the representation. Parametric representation fields transformed by neural functions support representing smooth continuous surfaces rather than disjoint patches. The implicit function values allow differentiable merging and transitions between representations. In addition, by optimizing neural representation fields to fit observed data, adaptive levels of detail can be obtained without extensive engineering of storage and resolution trade-offs. For large-scale reconstruction, there are many methods [7,8,9,10,11] leveraging grid- or voxel-based neural representations to accelerate computation and refine reconstructed details. But neural representations are typically global, so optimizing one parameter affects the entire scene, which makes it difficult to quickly process problems like loop closure.

Towards these problems, we aim to develop a novel surface representation like the papier-modeling method, while overcoming the aforementioned limitations. We introduce Neural Surfel Reconstruction, a framework that proposes the use of surfels (shown in Figure 2) as the basic geometric representation, augmented with neural descriptors to store shape information. Compared to grid- or voxel-based neural representations, surfel-based formulations inherently better preserve geometric structure. To equip surfels with the ability to represent complex and continuous shapes, a compact neural descriptor will be associated with each surfel, trained end-to-end from data using deep learning techniques. The neural descriptors provide powerful generalized features to describe complex surface properties.

During reconstruction, on one hand surfels can update their corresponding pose information in real-time through deformation graph optimization based on constraints like loop closures. On the other hand, the point cloud information contained in the surfels allows optimizing their neural features to obtain high-precision implicit representations or triangle meshes.

To demonstrate the advantages of the proposed surfel-based neural representation, we construct a system for online incremental 3D reconstruction from depth or point data. The system comprises three key components: (1) extraction of neural surfels from input data, (2) registration of new surfels into the incrementally built global graph, and (3) loop closure detection and pose optimization based on surfels. We evaluate our system extensively on various indoor and outdoor datasets. The results show that our approach can effectively combine the strengths of surfel and neural representation, enabling neural reconstruction systems to handle loop closure and bundle adjustment. In summary, the contributions of this work are the following:We propose Neural Surfel Reconstruction, a 3D neural reconstruction system with loop closure constraints, which first combines learning geometric neural features with surfel elements.We design a novel surface representation using new type of surfel equipped with neural descriptors that unify geometry and position for robust 3D reconstruction.We employ surfel pose graph optimization for our neural surfels to improve tasks such as scene reconstruction in environments with large loops.

## 2. Related Works

Three-dimensional reconstruction of real-world scenes remains a core objective in computer vision. This process entails capturing the shape and appearance of real-world objects and environments to generate digital 3D models. A variety of sensors and techniques are employed, each offering distinct advantages. However, these methods also possess inherent limitations.

### 2.1. Classical Geometry-Based Methods

A range of instruments, including stereo cameras, laser rangefinders, monocular cameras, and RGB-D cameras, are utilized to precisely perceive the three-dimensional world. Notably, advancements in consumer-grade RGB-D cameras have significantly propelled the development of visual scene reconstruction methods. These cameras provide aligned color and depth information at high frame rates, essential for crafting detailed 3D models. Using volumetric fusion techniques, such as those described in [2,12,13], depth maps are fused into a comprehensive 3D model. These techniques reconstruct surfaces by averaging truncated signed distance functions (TSDFs) [14] across a voxel grid, integrating multiple depth maps into an implicit surface that converges on the true surface.

An alternative method for constructing 3D models involves the use of point clouds, which have seen considerable advancements in point-based methodologies. Ref. [15] employs a straightforward, flat point-based representation that effectively manages larger spatial scales and dynamic scenes. Building on this, [16] utilizes anisotropic point representations along with memory-efficient attribute handling to efficiently reconstruct large-scale scenery. Further refining this approach, [17] focuses on minimizing registration errors by leveraging surface curvature as a dependable feature.

Conversely, surfel-based approaches maintain the global model using a set of surfels [18], which are circular discs characterized by positions and normals that are incrementally updated from new frames. This method circumvents the memory overhead associated with volumetric grids and reduces redundancy found in point cloud representations. Building on [15], ElasticFusion [2] integrates real-time loop closure handling similar to the techniques described in [3]. Furthermore, ref. [19] presents a probabilistic surfel map representation for Simultaneous Localization and Mapping (SLAM), utilizing a post-processing step to reconstruct a mesh from deformed keyframe depth maps. However, because each depth map is processed independently, this approach can result in the creation of multiple meshes for the same surface, leading to inconsistencies in the global map.

Unlike RGB-D cameras, which face illumination constraints in outdoor scenes, LiDAR scanners are used for large-scale mapping and directly provide sparser depth measurements in the form of 3D point clouds. This capability typically leads to LiDAR mapping approaches, such as LOAM [20], which represents the map as a point cloud. Their method achieves both low-drift and low-computational complexity without the necessity for high-accuracy ranging or inertial measurements. Meanwhile, [4] utilizes a surfel map to construct an efficient SLAM system for mapping three-dimensional laser range data, resulting in globally consistent maps. Ref. [21] improves camera poses in LiDAR maps using 2D–3D line correspondences. Others like [22,23] utilize point cloud to directly generate mesh as map representation.

However, these explicit representations generally discretize the scene at a fixed spatial resolution, which can be time-consuming and often lacks scalability. Furthermore, they struggle to make plausible geometric estimations for regions that remain unobserved.

### 2.2. Learning-Based Methods

Recent advances in neural implicit representation have shown significant advantages over the traditional explicit map representations that dominate current practices. Instead of directly storing attributes within a grid map, these methods use neural networks to approximate scene observations. Neural implicit map representations stand out by offering compact storage, improved noise smoothing capabilities, and enhanced inpainting and hole-filling in cases with sparse or occluded observations. Methods like [24,25] employ neural networks to regularize RGBD fusion amid noise and data gaps. Meanwhile, Droid-slam [26] integrates classical geometry with learned components in visual SLAM, resulting in significantly fewer catastrophic failures.

Beyond predicting intermediate 3D representations such as point clouds or voxels, several recent studies [1,27,28,29] investigate the direct generation of mesh representations from point clouds or images using neural networks. These mesh prediction techniques offer the benefit of more explicitly preserving the topology and structure of 3D surfaces.

Neural implicit representations such as DeepSDF [5] and IM-NET [30] have proven to be powerful tools for scene representation. These methods are capable of reconstructing surfaces and shapes by mapping coordinates to properties like distance or occupancy probability. Approaches such as [8,31] model scenes as continuous occupancy functions using neural networks, with representations based on a grid system. Moving beyond modeling entire scenes with a single multi-layer perceptron (MLP), newer techniques employ a hybrid representation that combines explicitly stored local latent features with a shallow MLP. For instance, LIG [11] leverages part-level geometric features using an overlapping latent grid to achieve high-fidelity scene representation. Similarly, POCO [32] attaches latent features directly to input points and utilizes point convolutions, maintaining a closer connection to the input surface.

End-to-end recurrent neural networks [33,34] are utilized for image inputs, capable of reconstructing 3D object shapes from single- or multi-view images and outputting a voxelized grid representation [35]. NeRF [36] introduces a radiance field to represent objects and scenes. Building on this, studies like [10,37,38,39,40,41] integrate depth information into the neural radiance field framework, using an implicit surface representation to depict scene geometry. These models update their implicit representations with various fusion algorithms, thereby enhancing the quality of 3D scene reconstruction and camera pose tracking. Additionally, the use of signed distance functions (SDF) has proven effective in scene reconstruction, with [42,43,44] delivering high-quality surfaces from multi-view images, combining the precision of implicit surface representations with the robust optimization techniques of neural radiance fields.

Furthermore, there are several methods based on intrinsic representation designed to model large-scale scenes [45,46]. Utilizing an RGB-D sequence, iMAP [6] introduces a real-time dense SLAM approach that employs a single multi-layer perceptron (MLP) to compactly represent the entire scene. Due to the limited capacity of a single MLP, iMAP struggles with producing detailed scene geometry and accurate camera tracking in larger scenes. While NICE-SLAM [9] introduces a real-time, scalable RGB-D SLAM system with a hierarchical, grid-based neural implicit encoding that enables local updates for large-scale environments. SurfelNeRF [47] is a work highly relevant to ours, which introduces the concept of surfel and combines it with implicit representation to obtain reconstruction results. Gaussian splatting-based methods [48,49], while effective for point-based rendering tasks, often fall short in capturing fine surface details, leading to blurred mesh reconstructions in complex scenarios. In contrast, our Neural Surfel Reconstruction method uses voxel-based storage and neural latent codes to ensure higher geometric accuracy and robust loop closure integration.

## 3. Neural Surfel Reconstruction

As shown in Figure 3, the system comprises three core modules: Neural Representation, Tracking and Mapping, as well as 3D reconstruction. First, the Tracking and Mapping module follows a traditional surfel-based SLAM pipeline [2,4] to extract surfels from new frames and incrementally fuse them with the existing Neural Surfel Pose Graph, pruning the graph from redundant surfels as needed.

Next, two steps are executed in parallel: (1) loop closure detection determines if loop closures or bundle adjustments are needed, running surfel pose graph optimization to update surfels’ pose information. In the (2) neural representation part, we apply signed distance functions (SDFs) for modeling surfaces and shapes. An SDF defines the shortest distance from the point in space to the surface boundary, whose sign indicates whether the point is inside or outside the surface. As illustrated in Figure 4, considering a surface *S*, the SDF function SDF(x) at a query point x is defined as follows:(1)$$SDF\left(x\right)=\left\{\begin{array}{cc} d(x,S), & \mathrm{if}\;x\;\mathrm{is}\;\mathrm{outside}\;S;\\ -d(x,S), & \mathrm{if}\;x\;\mathrm{is}\;\mathrm{inside}\;S. \end{array}\right.$$Here d(x,S) denotes the distance from the point to the closest surface of *S*. The surface *S* is defined as follows:(2)$$S=\{x\in R^{3}|SDF\left(x\right)=0\}.$$

Then, new incoming neural surfels have their predicted local SDF values incorporated into the updated map with new poses.

For reconstruction, the mesh is obtained by globally interpolating and marching cubes over the joint coordinate frame after blending the geometric and neural representations.

### 3.1. Neural Surfel Representation and Extraction

When an input frame is received, we extract several surfels from it. As shown in Figure 2b, a neural surfel refers to a cubic voxel space containing geometric information, which is considered to be a partial surface of an object. Formally, we represent each neural surfel $s_{i}$ as follows:(3)$$s_{i}=\left\{R_{i},t_{i},r_{i},z_{i},P_{i}\right\},$$
where $R_{i}\in {\mathrm{SO}}_{3}$ is the rotation, $t_{i}\in R^{3}$ is the position, $r_{i}$ refers to the size of bounding box, $z_{i}$ is the corresponding neural descriptor, and $P_{i}=\left\{p_{1},p_{2},\ldots ,p_{m}\right\}\in m\cdot R^{3}$ is the point cloud on the neural surfel.

It can happen that the cubic voxel space of each surfel contains multiple surfaces, which means
(4)$$\exists \;p\;\mathrm{inside}\;\mathrm{bounding}\;\mathrm{box}\;\mathrm{of}\;\mathrm{surfel}\;s_{i},\;p\notin P_{i}.$$

That is, we can not use just one piece of “papier” to represent this situation, so we need to split them to different neural surfels. During the surfel extraction process, the key elements of our system are the following steps:Randomly sample neural surfels densely from the original point cloud (at a rate of around 1:200–1000). This initializes an overcomplete set of surfels covering the object’s surfaces.Prune surfels that are too close to each other based on distance and normal deviation thresholds. This removes redundant surfels representing the same local region.For each remaining surfel, associate nearby points using K-Nearest Neighbor (KNN) search and generate an initial volumetric bounding region. Specifically, set a K value (e.g., K = 5) and calculate the voxel size as twice the distance between the surfel and its K-th nearest neighbor. This provides a spatial extent for the nearby points.Re-assign the points within each surfel’s volume to that surfel.Further cluster the points within each surfel’s volume into different surface patches and create one surfel per patch. Then, update those surfels’ associated points based on which cluster its center falls into.Outlier surfels that have a point cloud size of less than 20 points are removed, which reveals missing surfaces. New surfels will be initialized at these cluster centroids.

### 3.2. SDFs Prediction

To encode the points into the neural surfel and later decode them into an SDF, we use a multi-layer perceptron (MLP) as described in DeepSDF [5]. We train the MLP using point cloud data from SceneNet datasets [50], with their SDFs computed by traditional methods from the corresponding synthetic meshes.

To simplify the learning process and improve generalization capabilities, each neural surfel encompasses a cubic space where the point cloud is normalized to be within a range of ${\left[-1,1\right]}^{3}$. Specifically, for each surfel $s_{i}$, any local point $x\in {\left[-1,1\right]}^{3}$ in the corresponding space of $s_{i}$ would globally satisfy:(5)$$r_{i}\cdot \left(R_{i}^{-1}x+t_{i}^{-1}\right)\in {\left[-1,1\right]}^{3},$$
which makes the network focus more on local shape rather than absolute coordinate values, improving generalization to similar shapes.

We use a MLP model $f_{\theta }\left(z_{i},x\right)$ to predict the SDF value at any 3D point x, based on a neural descriptor $z_{i}$ that encodes local shape properties. The model has trainable parameters θ that are optimized during training. That is, for any local point x in the bounding box,
(6)$$f_{\theta }\left(z_{i},x\right)\approx SDF_{i}\left(x\right),$$
where $SDF_{i}\left(x\right)$ is the local SDF value. For global cases ($x\in R^{3}$), we have
(7)$$SDF\left(x\right)\approx \frac{1}{\left|\Omega (x)\right|}\sum_{s_{i}\in \Omega \left(x\right)}r_{i}\cdot f_{\theta }\left(z_{i},r_{i}\cdot \left(R_{i}^{-1}x+t_{i}^{-1}\right)\right),$$
where Ω(x) denotes a set of surfels whose cubic space contains the point x, and $r_{i}$ refers to the size of the bounding box, which is defined in Equation (Equation 3).

### 3.3. Training and Inference

During training, each surfel $s_{i}$ is associated with a SDF sample pair $\chi_{i}=\{(x_{k},SDF_{i}\left(x_{k}\right)$, $k=1,2,\ldots ,m\}\left(x_{k}\in {\left[-1,1\right]}^{3}\right)$ capturing local shape information. We then optimize the network parameters θ and the neural descriptor $z_{i}$ for each surfel sample.

Thus, for every training pair $\left(s_{i},\chi_{i}\right)$, we have one SDF prediction $f_{\theta }\left(z_{i},x\right)$. Then, the loss function is
(8)$$L\left(s_{i},x\right)=\left|f_{\theta }\left(z_{i},x\right)-SDF_{i}\left(x\right)\right|.$$

After minimizing the negative log posterior, the optimization results are as follows:(9)$$\arg\min_{\theta ,\{z_{i}\}}\sum_{i=1}^{n}\left(\sum_{x_{k}\in \chi_{i}}L\left(s_{i},x_{k}\right)+\frac{1}{\sigma^{2}}{∥z_{i}∥}_{2}^{2}\right).$$

For inference on novel partial shapes, the neural descriptors are initialized randomly and then optimized to reconstruct the shape by minimizing the loss between predicted SDF values and observations on the given points. That is, the weights θ are fixed and the neural descriptor $z_{i}$ will be found by
(10)$$\arg\min_{z_{i}}\sum_{x_{k}\in \chi_{i}}L\left(s_{i},x_{k}\right)+\frac{1}{\sigma^{2}}{∥z_{i}∥}_{2}^{2}.$$

The previously demonstrated pipeline is offline to facilitate graph optimization. However, we can still achieve online reconstruction by using a neural surfel representation as a substitute for the traditional surfels of ElasticFusion. During the mapping process, we can query the SDF value at any position using the global neural surfel map, enabling mesh reconstruction through the marching cubes algorithm [51].

### 3.4. Neural Surfel Pose Graph

Neural surfel pose graph optimization is introduced from [2,3,52,53], which is designed for shape manipulation of mesh and loop closure cases of traditional surfels. Specifically, the surfel pose graph enables non-rigid registration between different parts of the incrementally built surfel model to achieve loop closure and global consistency. In our approach, the graph G=(V,E) will be defined by
(11)$$\begin{array}{cc} V=\{ & s_{i},i=1,2,\ldots ,n\},\\ E=\{ & \left(s_{a},s_{b}\right)|\exists \;p\in P_{a}\cap P_{b}\\ & \;\mathrm{or}\;s_{a}\;\mathrm{and}\;s_{b}\;\mathrm{are}\;\mathrm{in}\;\mathrm{the}\;\mathrm{same}\;\mathrm{frame}\;\}. \end{array}$$

All neural surfels are the graph nodes, and if two surfels share the same point or originate from the same frame of data, there exists an edge between them.

For the nearby space of each surfel $s_{i}$, when assigning an affine transformation $\left(T_{R}i,T_{t}i\right)$, we have the affine transform of node *j* for the point v: (12)$$A_{i}\left(v\right)=\tilde{v}=T_{R}i\left(v-t_{i}\right)+T_{t}i+t_{i}.$$

Because the deformation is determined by multiple surfels, in the global case, the deformed position of each surfel $s_{i}$ is
(13)$$\phi \left(t_{i}\right)=\sum_{j\in \Pi (i)}w_{j}\left(t_{i}\right)A_{i}\left(t_{i}\right),$$
where Π(i) denotes the set of connected surfels of $s_{i}$, and the weights $w_{j}\left(\cdot \right)$ linearly fall with increasing distance to the surfel, and are then normalized to sum to 1.

The optimization minimizes an objective function with several terms. The rotation energy minimizes the difference between each node’s affine transformation and a true rotation matrix. Specifically, it calculates the Frobenius norm between each rotation matrix: (14)$$E_{rot}=\sum_{j=1}^{N}{∥T_{R}j^{⊤}T_{R}j-I∥}_{F}^{2},$$
where I is the identity matrix. If a loop exists, the surfel will be updated with its new pose, which contains a destination position $q_{i}$. Then the position energy incorporates user position constraints into the objective. It sums the squared error between the predicted transformed position and specified target for each constraint point:(15)$$E_{pos}=\sum_{i=1}^{N}{∥\phi \left(t_{i}\right)-q_{i}∥}_{2}^{2}.$$The regularization term minimizes differences between transformations of neighboring nodes for smoothness. It computes the deviation between each node’s transformation applied to its neighbors and their actual transformed positions:(16)$$E_{reg}=\sum_{j=1}^{N}\sum_{k\in \Pi (j)}{∥A_{j}\left(t_{k}\right)-A_{k}\left(t_{k}\right)∥}_{2}^{2}.$$

Thus the final energy is
(17)$$E=w_{rot}E_{rot}+w_{pos}E_{pos}+w_{reg}E_{reg}.$$

### 3.5. Mesh Generation

After optimizing the neural surfels’ SDF predictions to fit the observed data, we can generate a mesh reconstruction by densely querying the SDF values and extracting an isosurface by Equation (Equation 7).

However, this can lead to ambiguity in SDFs of certain points, because two neural surfels that are not connected to each other can still have overlapping regions in their corresponding cubic spaces. Figure 5 illustrates this situation, and we will set the threshold *T* to truncate points with large signed distance values, which means
(18)$$TSDF\left(x\right)=\left\{\begin{array}{cc} SDF(x), & \mathrm{if}\;SDF(x)\leq T;\\ none, & o.w. \end{array}\right.$$

For practicality we employ a simpler approach. The TSDF predictions are directly obtained and mapped to the global representation. Then, re-sampling and interpolation is conducted at the global level to obtain a TSDF on a regular grid. Finally, the mesh is generated from the global TSDF using techniques such as marching cubes.

## 4. Results and Evaluation

### 4.1. Data Preparation

Before training the MLP for neural surfel representation, we begin with watertight synthetic objects [54] that have a mesh representation. For each mesh, we start by performing subdivision to obtain a higher density of mesh vertices. From these vertices, we randomly select one as a surfel $s_{i}$ and set the corresponding a random number $r_{i}$ to decide the cubic space. We then choose connected mesh vertices of this surfel and relative triangles to obtain the ground truth SDF, and sample the point cloud $P_{i}$ on this partial mesh.

For testing on one surfel, we first compute the normal $n_{j}$ of the point $p_{j}\in P_{i}$. For each view frame $f_{j}$’s pose, we will decide whether to update the direction of normal $n_{j}$ based on the angle between the view direction to the point positions in $p_{j}$ and the current normal $n_{j}$: if this angle is within 180 degrees, we keep the current normal, otherwise the normal direction is updated to face the view direction. The initial SDF values are computed based on $P_{i}$, by finding its nearest neighbors in the point cloud and calculating distance along the normal direction.

### 4.2. Surfel Graph Optimization

We evaluate the ability of the surfel graph deformation framework to accurately register and close loops during incremental scanning. Since SceneNet [50] and Replica [55] are synthetic or high-quality datasets, we selected eight rooms to test our system. Figure 6a involves loop closure of a simulated circular scan containing 1261 surfels, which adds noise to translation term for each surfel. We compare all surfels’ positions and the ground truths.

Similar with absolute trajectory error (ATE), we use the absolute surfel position error (ASE) to evaluation our optimization.
(19)$$ASE=\sqrt{\frac{1}{n}\sum_{i=1}^{n}{∥t_{i}-{\hat{t}}_{i}∥}^{2}},$$
where $t_{i}$ is the position of optimized surfel $s_{i}$, and the ${\hat{t}}_{i}$ is the ground truth. Figure 7 and Table 1 show the evaluation about deformation graph optimization, which demonstrates that the graph optimization can effectively achieve loop closure by non-rigid alignment of surfels. The node connectivity based on shared points successfully prevents distorting uncorrelated surfaces. The combination of local loop closure and global pose graph constraints maintains globally consistent models.

### 4.3. Object Reconstruction

For object reconstruction cases, we assume the poses of all surfels to be correct, so the deformation graph would not be constructed. We evaluate the neural surfel representation on object-level shape reconstruction, and point cloud inputs (4000 points) are generated by sampling from the original mesh.

A quantitative comparison with DeepSDF [5] and Deep Local Shapes (DeepLS) [7] is shown in Figure 8 and Table 2. We report mean chamfer distances between reconstructed meshes and ground truth. Overall, the results of Neural Surfel Reconstruction are comparable to those of DeepSDF. However, because the complexity of the object surfaces is generally relatively low, we randomly sampled 10 points as initial surfel positions for each object. As a result, each object ended up having 7∼13 surfels. This approach did not yield a significant advantage in terms of capturing finer details. Because DeepLS uses over a hundred regularized blocks, the edges inevitably fit closely with the synthetic mesh data.

### 4.4. Scene Reconstruction

Finally, we showcase a full scene reconstruction by scanning several scenes containing multiple surfaces. We utilized several samples from the SceneNet [50], ScanNet [56], Dyson Lab [2], Replica [55], and our UAV data as our test data for scene reconstruction. For synthetic data, like SceneNet, we simulate a user performing a looped scan holding an RGB-D sensor, generating several frames of input point cloud, as shown in Figure 6. For real data, we follow the real data scanning process to reconstruct the scene and test both RGB-D and LiDAR point cloud cases.

The comparison, illustrated in Figure 9 and Table 3, includes DeepSDF [5], DeepLS [7], ConvONet [8], POCO [32], LIG [11], and Poisson [22] (with an octree depth value of 8). The input point clouds for DeepSDF, DeepLS, POCO, and LIG were obtained using KISS-ICP [57], which does not include loop closure constrains. For other methods, the point clouds and surfel information were acquired using ElasticFusion [2]. Poisson* denotes that we retained the surfel structure, and each surfel’s surface was reconstructed using the Poisson reconstruction method. Thus, in the Poisson* results, it appears that multiple layers of meshes are stacked together.

We found that our results can accurately capture fine details, and thanks to the surfel graph structure design, Poisson* can even also effectively represent the detailed aspects of the scene. In those cases with acceptable drift, such as ScanNet, POCO and LIG sometimes exhibit rendering errors similar to pose misalignment. While their methods capture more detail than ours, this highlights an area for future improvement in our approach. In scenarios where the UAV performs long-range scans, the use of GPS for global positioning allows almost all methods to accurately capture details. In this setting, our results still demonstrate a comparable reconstruction result. Since DeepSDF and DeepLS are specifically designed for watertight object-level mesh reconstruction, they perform poorly on large-scale data, as shown in Figure 10.

In addition, we have compressed the representation of the entire scene by only preserving necessary information $\left\{R_{i},t_{i},r_{i},z_{i}\right\}$ about the surfels, and the comparison with initial point clouds is shown by Table 4. We found that our method produces smaller file sizes compared to directly reconstructing the entire point cloud using Poisson at a low level.

### 4.5. Failure Cases and Limitation

Figure 5b demonstrates an issue during reconstruction where additional surfaces are erroneously added to certain edge surfaces. In the right surfel of the graph, SDF values actually span across the x-y plane, resulting in the incorrect retention of extra surfaces. Reconstruction of large-scale scenes may also exhibit surface aliasing. However, in the case of object reconstruction, where the point cloud is typically continuous (similar to directly sampling from a watertight mesh), this issue would not occur. One solution is to refine the surfels by decreasing the range of the cubic space for each surfel, thus reducing the occurrence of such situations.

Neural Surfel Reconstruction is a reconstruction system; however, it has certain limitations. One significant limitation is that it cannot run in real time. The computational demands of the system, particularly the learning-based algorithms and the graph-based optimization processes, require substantial processing power and time. As a result, this system is better suited for offline processing rather than real-time applications. Consequently, scenarios that require immediate feedback or real-time interaction, such as live robotic navigation or interactive AR/VR environments, may not benefit from this approach. Future work could focus on optimizing the computational efficiency to bring the system closer to real-time performance.

## 5. Conclusions

In this work, we have presented *Neural Surfel Reconstruction*, a novel representation combining surfels and neural features. Surfels act as nodes in a pose graph for loop closure while attaching a neural shape descriptor, which preserves geometric structure and enables efficient non-rigid registration. SDF predictions are blended from local surfels to generate a coherent and detailed surface representation. This approach leverages the strengths of both local and global information, allowing for more accurate and continuous surface reconstruction. Our experiments demonstrate that this hybrid approach provides good geometric structure with neural representation, effectively capturing detailed surface information and ensuring robust reconstruction across various scenarios. In the future, we plan to enhance the neural descriptors and optimize the computational efficiency. This surfel-based neural representation improves scalability and efficiency of learning-based reconstruction. We believe it paves the way for real-time neural point-based SLAM. 

## Figures and Tables

**Figure 1 sensors-24-06919-f001:**
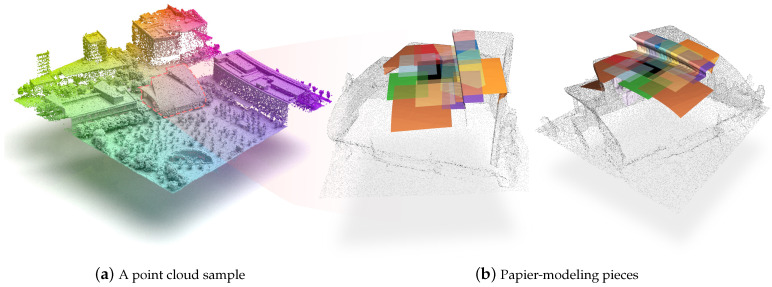
Teaser image: Schematic diagram of the paper-modeling method. We mounted a LiDAR scanner on the UAV to capture point cloud data (**a**) of an area. In (**b**), each color represents a piece of a papier, and the fusion of all these papiers is a portion of the object’s surface.

**Figure 2 sensors-24-06919-f002:**
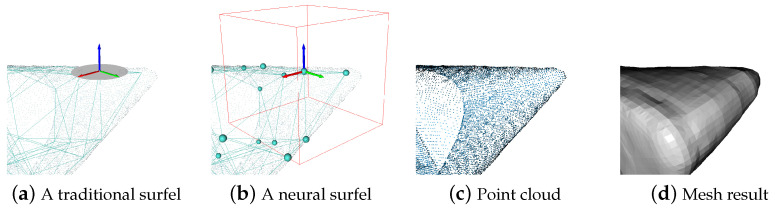
Compared to traditional surfels, our proposed neural surfels replace the basic disk (gray disk in the (**a**)) representation with a cubic voxel space (red box in the (**b**)) to store richer geometric information. Based on the point cloud within the voxel, our approach selects the coordinates of the point cloud that are connected to the surfel to learn neural latent codes and predict SDF values. Moreover, in traditional surfel reconstruction, not all surfels are used to constitute a deformation graph. However, in the definition of neural surfels reconstruction, all surfels are considered as graph nodes. (**c**) is the input point cloud, and (**d**) is the mesh result generated by our system.

**Figure 3 sensors-24-06919-f003:**
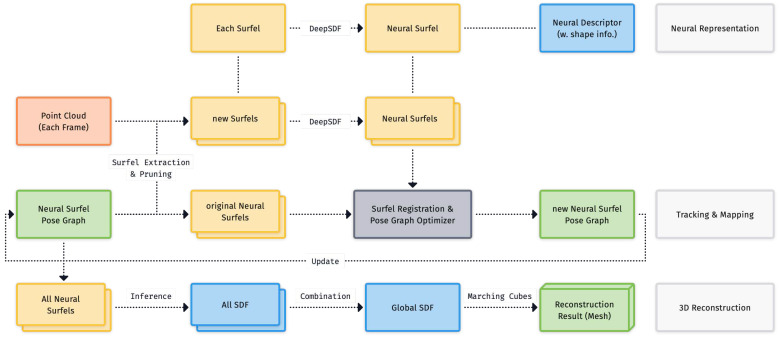
The system consists of three core modules: Neural Representation, Tracking and Mapping, as well as 3D reconstruction. First, we extract the neural surfel with necessary pruning (refer to Section 3.1). Each surfel comes with a specified neural descriptor to represent its particular shape (refer to Section 3.2 and Section 3.3). Following this, the surfel pose graph optimization module is used (refer to Section 3.4) to update the graph. Finally, all surfels’ SDF are fused to produce the final reconstruction mesh (refer to Section 3.5).

**Figure 4 sensors-24-06919-f004:**
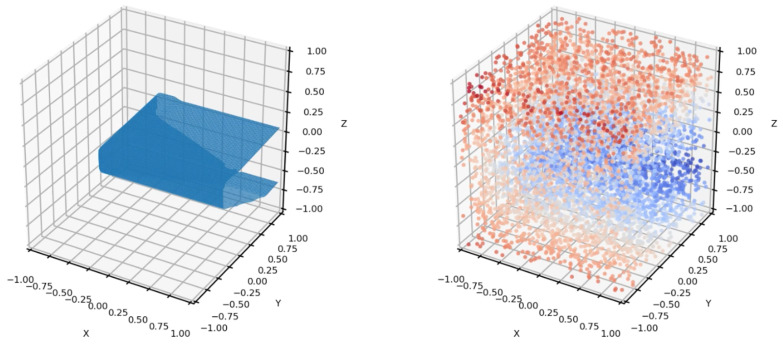
Illustration of the cubic space of a typical surfel, showing both the mesh representation and the sampled signed distance function (SDF) values. In the SDF, blue indicates negative distances inside the shape, while red indicates positive distances outside.

**Figure 5 sensors-24-06919-f005:**
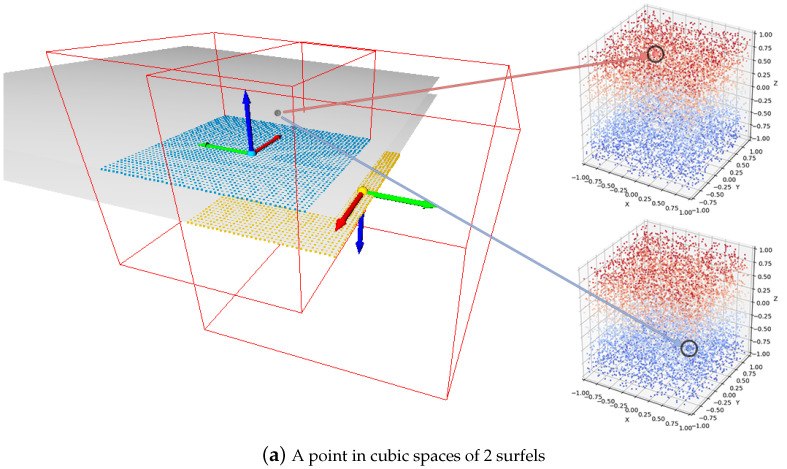
A point can be present in two surfel cubic spaces that are not neighbors, resulting in outliers. There are many such cases in our system, such as corners or on the upper and lower surface of a table. In (**a**), this point is positive (red) for the upper surface and negative (blue) for the bottom surface. To address this issue, we make use of the truncated signed distance function (TSDF) to truncate points with large signed distance values. Illustrated by (**b**,**c**), this approach is based on the assumption that distances on all surfaces are larger than 2T, where *T* is a specified threshold.

**Figure 6 sensors-24-06919-f006:**
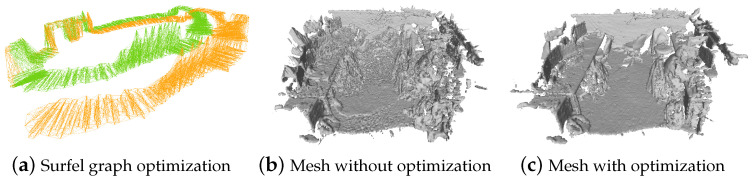
A case about scene reconstruction with loop closure. (**a**) shows the initial graph (orange) and the deformed graph (green) after optimization, and (**c**) is the final reconstructed mesh with global consistency.

**Figure 7 sensors-24-06919-f007:**
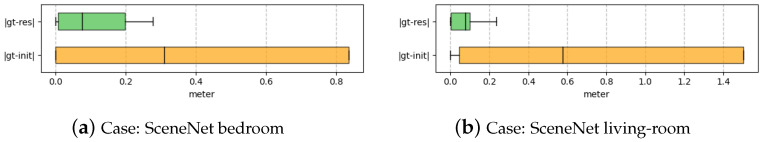
The optimization results for neural surfel graph. It shows the absolute distance $\left|t_{i}-{\hat{t}}_{i}\right|$ between the test data and ground truth. The green one is our method, and the orange one is the initial surfel with added noises.

**Figure 8 sensors-24-06919-f008:**
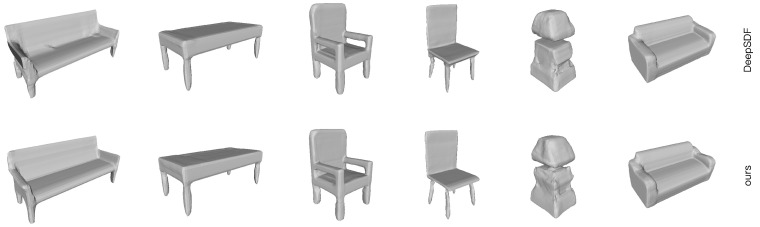
Qualitative comparison of our method with DeepSDF.

**Figure 9 sensors-24-06919-f009:**
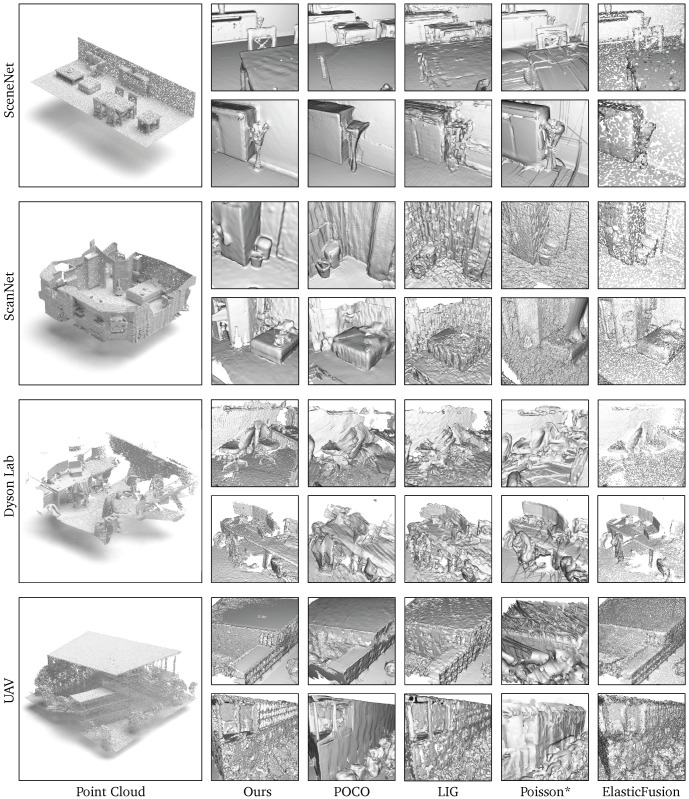
Qualitative comparison between our Neural Surfel Reconstruction, POCO, LIG, Poisson*, ElasticFusion on the various datasets. Here, Poisson* denotes Poisson-based surfel reconstruction, where we replace our neural representation module with the Poisson method.

**Figure 10 sensors-24-06919-f010:**
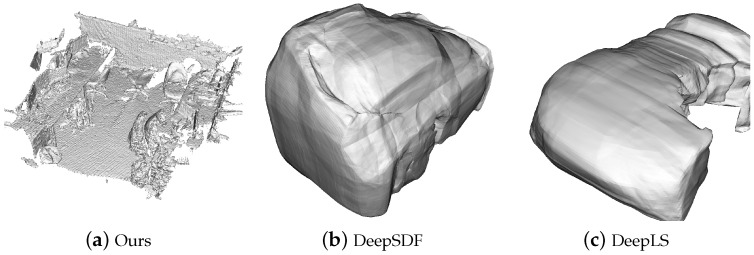
Qualitative comparison between our Neural Surfel Reconstruction, DeepSDF and DeepLS on Dyson Lab data.

**Table 1 sensors-24-06919-t001:** The optimization results for deformation graph. It shows the ASE between the test data and ground truth.

ASE (m)	Initial Position	Optimized Result
SceneNet Bedroom	0.9781	0.0809
SceneNet Living-room	0.5323	0.1341
SceneNet Office	0.7122	0.0731
Replica Office 2	0.6441	0.1202
Replica Office 3	0.6231	0.0911
Replica Apartment 0	1.2021	0.3218
Replica Room 0	0.4230	0.0208
Replica Room 1	0.7413	0.0730

**Table 2 sensors-24-06919-t002:** Reconstruction performance on ShapeNet dataset. It shows the chamfer distances (CD) between the reconstructed meshes and ground truth.

CD (↓)	Ours	DeepSDF	DeepLS
sofa	0.132	0.141	0.044
chair	0.204	0.117	0.030
lamp	0.832	1.034	0.078
table	0.553	0.341	0.032

**Table 3 sensors-24-06919-t003:** Reconstruction performance on SceneNet dataset.

Method	SceneNet [50]			Replica [55]		
	CD (↓)	Normal (↑)	F-Score (↑)	CD (↓)	Normal (↑)	F-Score (↑)
DeepSDF [5]	4.611	0.510	0.068	8.123	0.122	0.232
DeepLS [7]	12.836	0.001	0.470	5.642	0.041	0.341
ConvONet [8]	0.076	0.510	0.692	0.082	0.412	0.703
LIG [11]	0.059	0.517	0.623	0.043	0.519	0.663
POCO [32]	0.062	0.547	0.652	0.041	0.621	0.688
Poisson	0.084	0.374	0.401	0.120	0.476	0.612
Ours	0.056	0.578	0.694	0.048	0.682	0.692
Poisson*	0.049	0.445	0.854	0.086	0.511	0.820

**Table 4 sensors-24-06919-t004:** The storing space for our Neural Surfel Reconstruction. Here, Poisson denotes that we used the Poisson method (with an octree depth value of 8) to directly reconstruct the entire point cloud, and Poisson* denotes Poisson-based surfel reconstruction.

File Size (MB)	Point Clouds	Ours	Poisson	Poisson*
SceneNet	132.8	2.8	4.8	1224.3
ScanNet++	42.9	7.1	12.3	612.2
Dyson Lab	28.0	0.8	10.6	295.6
UAV	411.7	3.2	34.9	3422.1

## Data Availability

No new indoor data were created or analyzed in this study, the UAV point cloud in this work will be available online at http://jdtsui.github.io/neural-surfel/ (accessed on 3 July 2024).

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
