# Peer review of "Neural Surfel Reconstruction: Addressing Loop Closure Challenges in Large-Scale 3D Neural Scene Mapping"

_sensors, 2024, doi:10.3390/s24216919_

Round 1
Reviewer 1 Report
Comments and Suggestions for Authors
My main concern with the paper is the lack of even mention of Gaussian splatting, as explained above. It needs to be addressed in some form - it would be sufficient to explain why it is inappropriate to compare to it, or to point out at least one major way in which it is inferior, e.g. runtime, flexibility for integrating loop closure, etc. If there is no such feature in which neural surfels are better, the scientific contribution of this work significantly decreases.
Comments on the Quality of English LanguageNothing too major. E.g. looking at the first sentence "implicit neural representation is a popular topic in 3D surface reconstruction" - implicit neural representations would be correct. So I'd recommend proof-reading for such grammatical mistakes. In addition, make sure there is a space beforethe brackets [] for a citation, e.g. "Droidslam[25]" needs to be "Droidslam [25]".
Reviewer 2 Report
Comments and Suggestions for Authors
This manuscript presents a neural surface reconstruction algorithm that can handle loop closure and bundle adjustment through the concept of surfel.
Although the method demonstrated in this manuscript simplifies the 3D reconstruction problem in loop closure, it still needs further improvements.
The manuscript contains some grammatical errors, and two of them are below. Please check them carefully.
1. This manuscript mistakenly uses the word 'paper', which should be 'papier' from [1].
2. A '.' is missed before 'In the (2)' in Line 169.
There is also some information that needs to be included in the manuscript.
1. There needs to be details about how to decide the size of the voxel in Eq. 3 and how to further cluster the points within each surfel.
2. Table 3 does not provide results of the Poisson and the Elastic Fusion.
3. In Figure 10, the results of DeepSDF and DeepLS can be improved by removing the ceiling part of the room.
In the end, the design of the experiments is confusing.
The Object Reconstruction Section is relevant to the proposed method, which is even worse than DeepSDF and DeepLS on several metrics. It would be better to focus on the proposed method's contribution.
In the Scene Reconstruction, it is unfair for the author to compare their method with DeepSDF, DeepLS, POCO, and LIG, as these methods are post-processing methods instead of incrementally reconstructing the scene. It would be better to compare which ElaticFusion, BundleFusion, or other related methods.
In the Surfel Graph Optimization, the author compares all surfel's initial position and optimized position in Table 1 and provides reconstruction results in Figure 6. However, there are no quantitative results of these results. Besides, it would be better to provide results from the Dyson Lab, UAV, and the ScanNet.
Comments on the Quality of English LanguagePlease check the manuscript carefully before resubmitting.
Reviewer 3 Report
Comments and Suggestions for Authors
1. What methods were used to combines learning geometric neural features with surfel elements? What are the important results?
2. Please include specific and quantitative results in the Abstract, while ensuring that it is suitable for a broad audience.
3. The importance of the problem should be given in Abstract section.
4. There is selected 2 rooms to test the system that the ability of the surfel graph deformation framework. The result may not be universal.
5. “We found that our method produces smaller file sizes compared to directly reconstructing the entire point cloud using Poisson at a low level. ” The language used in this article is not objective enough.
Average.
Round 2
Reviewer 3 Report
Comments and Suggestions for Authors
Minor editing of English language required.
Comments on the Quality of English LanguageMinor editing of English language required.
Author Response
Dear Editor,
thank you for giving us the change to further improve our paper by fixing some of the English. My advisor Prof. Schwertfeger, whose English is excellent, has proof read the paper and corrected some sentences.